# On Mechanical Properties of Welded Joint in Novel High-Mn Cryogenic Steel in Terms of Microstructural Evolution and Solute Segregation

**Jia-Kuan Ren** , **Qi-Yuan Chen, Jun Chen * and Zhen-Yu Liu ***

State Key Laboratory of Rolling and Automation, Northeastern University, Shenyang 110819, China; 18233561905@163.com (J.-K.R.); 13940584639@163.com (Q.-Y.C.)

* Correspondence: chenjun@mail.neu.edu.cn (J.C.); zyliu@mail.neu.edu.cn (Z.-Y.L.);
  Tel.: +86-024-8368-0571 (J.C.); Fax: +86-024-2390-6472 (J.C.)

**Abstract:** There is a growing demand for high-manganese wide heavy steel plate with excellent welding performance for liquefied natural gas (LNG) tank building. However, studies on welding of high-Mn austenitic steel have mainly focused on the applications of automotive industry for a long time. In the present work, a high-Mn cryogenic steel was welded by multi-pass Shielded Metal Arc Welding (SMAW), and the microstructural evolution, solute segregation and its effect on the properties of welded joint (WJ) were studied. The yield strength, tensile strength and elongation of the WJ reached 804 MPa, 1027 MPa and 11.2% at −196 °C, respectively. The elongation of WJ was reduced with respect to the BM due to the poorer strain hardening capacity of weld metal (WM) at −196 °C. The WM and coarse-grained heat affected zone (CGHAZ) had the lowest cryogenic impact absorbed energy of ~55 J (at −196 °C). The inhibited twin formation caused by the higher critical resolved shear twinning stress ($\tau_T$) in the C-Mn-Si segregation band, the inhomogeneous microstructure caused by solute segregation, and the hardened austenite matrix deteriorated the plastic deformation capacity, finally resulting in the decreased cryogenic impact toughness of the CGHAZ. To summarize, the cryogenic toughness and tensile properties of the WJ meet the requirements for LNG tank building.

**Keywords:** high-Mn austenitic steel; SMAW; cryogenic impact toughness; solute segregation; twinning

---

## 1. Introduction

With the rapid development of clean energy, there is an increasing demand for cryogenic vessel steels used in LNG (liquefied natural gas) storage tanks [1,2]. However, there exist some disadvantages for conventional cryogenic materials, such as high cost and poor welding performance [3,4]. Therefore, in searching for novel cryogenic materials, high-manganese austenitic steel becomes a potential candidate cryogenic material [5] due to its high cryogenic impact toughness [6,7] and ductility [8], good fatigue performance, etc. However, compared with ferritic steels, austenitic steels always possess lower thermal conductivity, higher thermal expansion coefficient and greater electrical resistance. Hence, the trends of grain coarsening, embrittlement and hot cracking in the heat-affected zone (HAZ) are relatively serious for high-manganese austenitic steels [9,10], resulting in poor cryogenic toughness in the HAZ.

The inhomogeneous microstructure always formed during welding plays an important role in mechanical properties [11]. The segregation of Mn and Mo at sub-grain boundaries lowers the critical resolved shear stress of the dendritic sub-grain area, thereby resulting in less energy absorption of crack propagation along the major axis of dendritic sub-grain in 316LN stainless steel [12]. Saeidi et al. [13]

---

also indicated that the compositional fluctuation due to the local enrichment of Mo induced a fine network of dislocation tangling, which would lead to locally hardened austenitic grains. The study by Xiao et al. [14] demonstrated that the solute segregation of austenitic stainless steel increased with the increasing heat input, which promoted the inhomogeneous plastic deformation, finally resulting in the deterioration of cryogenic impact toughness. These results suggest that the solute segregation is an important factor affecting the quality of welded joint (WJ) in austenitic steel. For a long time, studies on high-Mn austenitic have steel mainly focused on the applications of automotive industry for its excellent strain hardening capability [15]. Consequently, studies on welding of high-Mn austenitic steel are mainly related to electrical resistance welding (ERW) [16], laser welding [17] and friction stir spot welding (FSSW) [18], which have a small and concentrated welding heat input. Recently, there has been an increasing number of studies focusing on fusion welding of high-Mn wide heavy plates due to its growing demand for LNG tank building. Fusion welding processes with a high energy input are usually accompanied by manganese evaporation, dilution, segregation and co-segregation, which result in significant changes to the material in the fusion zone and the HAZ. Recent studies on fusion welding performance of high-Mn wide heavy plates have mainly focused on hot ductility behaviors [19,20] and hot cracking susceptibility [21], whereas the correlation between inhomogeneous microstructure in HAZ and cryogenic mechanical properties is still unclear.

The present work aims at elaborating the microstructural evolution, alloying element segregation and depletion in coarse grained heat-affected zone (CGHAZ) and their effects on cryogenic impact toughness. Additionally, the cryogenic impact toughening mechanism in CGHAZ is well clarified. This work is expected to be helpful for the engineering application of high-Mn cryogenic steels. The electron probe microanalyzer (EPMA) and wavelength dispersive spectrometer (WDS) were applied to investigate the alloying element segregation and depletion in CGHAZ and base metal (BM). The dependence of mechanical nano-twins on grain size and solute segregation under cryogenic-temperature dynamic loading conditions were characterized by means of transmission electron microscopy (TEM) and scanning electron microscopy (SEM).

## 2. Experimental Procedure

### 2.1. Materials

The chemical composition of the investigated high-Mn austenitic steel is presented in Table 1, which were prepared in high-frequency vacuum induction melting and molded cast. The cast ingot was homogenized at 1200 °C for 3 h and then hot rolled in seven passes to 12 mm at a temperature range of 1150–930 °C, followed by water cooling to room temperature.

**Table 1.** Chemical composition of the base material and filler metal in wt.%.

| Alloy | C | Mn | Si | Al | V | Cr | Mo | P | S | Fe | Ni |
|---|---|---|---|---|---|---|---|---|---|---|---|
| Base material | 0.57 | 24.4 | 0.48 | 1.92 | 0.31 | – | – | 0.004 | 0.005 | Bal. | – |
| Filler metal | 0.01 | 3.7 | 0.07 | – | – | 13.27 | 6.16 | 0.001 | 0.001 | – | Bal. |

### 2.2. Welding Procedure

Double-sided multi-pass Shielded Metal Arc Welding (SMAW) was performed using a new type of Ni-based electrode developed by Central Iron & Steel Research Institute. The diameter of core and the thickness of coat are 3.2 mm and 1.2 mm, respectively. In addition, the chemical composition of the core is presented in Table 1 and its mechanical properties are listed in Table 2. The high-Mn steel was cleaned and asymmetrical double-V grooves were machined, as shown in Figure 1. The dimension of welded plates is 600 mm (L) × 220 mm (W) × 12 mm (H), and the welding direction is parallel to the rolling direction. An arc voltage of 26–28 V, a welding current of 100–110 A, a welding speed of 19.8 cm·min$^{-1}$, an inter-pass temperature below 150 °C, a heat input of ~9 kJ·cm$^{-1}$ and welding passes of 13 were used during the SMAW process.

**Table 2.** Mechanical properties of welding core.

| Alloy | Yield Strength, MPa | Tensile Strength, MPa | Charpy V-notch Impact Energy, *J* (−196 °C) |
|---|---|---|---|
| Nickel-based welding core | 445 | 700 | 100 |

### 2.3. Mechanical Property Tests

Cylindrical tensile samples (the gage section Ø6 mm × 60 mm, orientation; transverse direction) were machined from welded plate and hot-rolled plate, and the weld metal (WM) was in the middle of the gage section. The hot-rolled plate, i.e., the base metal, was designated as "BM". The tensile tests were conducted on an AG-X plus pc-controlled tensile machine (Shimadzu Co. Ltd., Kyoto, Japan) at room temperature and cryogenic temperature (−196 °C), with a constant cross speed of 5 mm/min. During the cryogenic temperature tensile test, a low-temperature chamber containing liquid nitrogen was attached to the test machine. The 0.2% offset stress was determined to be the yield strength.

Standard Charpy V-notch impact samples (size; 10 mm × 10 mm × 55 mm, orientation; transverse direction) were machined from the thickness center of welded plates, as shown in Figure 1. To ensure the accuracy of Charpy V-notch locations, samples were macro-etched in a 4% nitric acid alcohol solution to visualize the HAZ outline. In accordance with the International Code of Safety for Ships Using Gases or Other Low-Flashpoint Fuels (IGF Code) 16.2.2.3 [22], Charpy V-notches were prepared at WM, fusion line (FL), and 1 mm, 2 mm, 4 mm, 5 mm and 25 mm from FL (FL + 1 mm, 2 mm, 4 mm, 5 mm and 25 mm) to test cryogenic impact toughness at these locations, as shown in Figure 1. FL is defined as the location where half of the V-notch is in WM [23]. Charpy impact tests were conducted on a pendulum impact testing machine (MTS systems Co. Ltd., Minneapolis, MN, USA) at −196 °C. To ensure accuracy, three tensile and impact samples were machined from each steel and tested in order to reduce error. Vickers macrohardness tests of WJ were conducted on KB3000BVRZ-SA hardness tester (KB Co. Ltd., Hamburg, GER) with 100 g load.

### 2.4. Microstructure Characterization

Microstructure characterizations of HAZ and BM were conducted on a Zeiss Ultra 55 (Carl Zeiss AG, Jena, Germany) scanning electron microscopy (SEM). The EBSD specimens were machined from the welded plate along the transverse section perpendicular to the welding direction, then polished and electro-polished for 27 s at 27 V at room temperature using a 12.5 vol.% perchloric acid alcohol electrolyte. The locations characterized by EBSD are indicated by red four-pointed stars in Figure 1. The EBSD specimens were detected using a scanning step of 0.45 µm and the EBSD data were post processed by HKL CHANNEL 5 software (version; 5.0.9.0, OXIG Co. Ltd., Oxford, UK). The optical microstructure (OM) specimens were characterized by OLYMPUS BX53 metalloscope (OLYMPUS Inc., Kyoto, Japan). The TEM specimens were cut along the rolling surface of the deformed regions close to fracture surface of Charpy V-notch impact samples where the Charpy V-notches were prepared at FL location and BM, and then mechanically thinned to 50 µm thickness. The TEM specimens with 3 mm diameter were punched from the thin foils and then further thinned by a Struers TenuPol-5 twin-jet electropolisher (Struers Inc., Copenhagen, Denmark) under 30 V at −25 °C in a 10 vol.% perchloric acid alcohol electrolyte, and then examined in a FEI Tecnai G2 F20 field-emission transmission electron microscopy (FEI Co. Ltd., Hillsboro, OR, USA) operated at 200 kV. To characterize the distribution of alloying elements in CGHAZ and BM, specimens were conducted in a JXA-8530F (JEOL Co. Ltd., Kyoto, Japan) electron probe microanalyzer (EPMA) at an operating voltage of 20 kV, a current of $2 \times 10^{-8}$ A and a step size of 0.5 µm. The location characterized by EPMA in CGHAZ are indicated by a pink four-pointed star in Figure 1. Fractographs of Charpy impact samples were also observed using SEM.

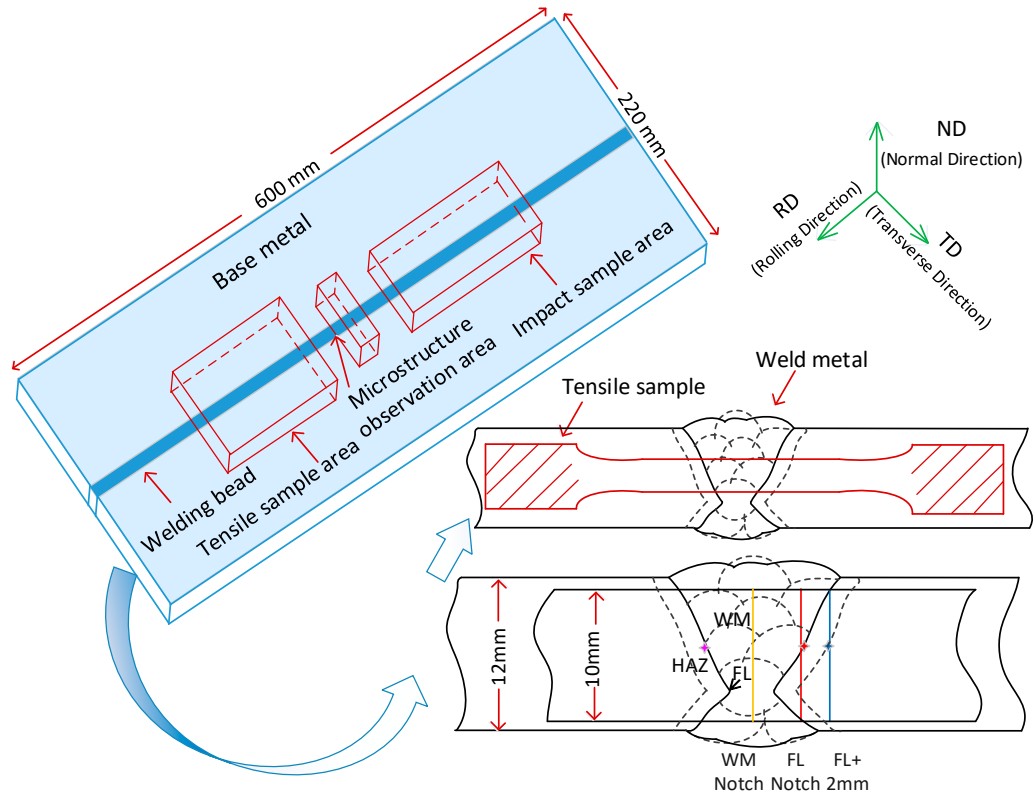

**Figure 1.** Schematic diagram of sampling location, welding groove, locations of partial Charpy V-notch (indicated by the yellow, red, and blue solid lines) and microstructure characterization locations (indicated by the red, blue and pink four-pointed star).

## 3. Results

### 3.1. Microstructure Characteristics

The optical microstructure of the WJ at FL, FL + 2 mm and BM is shown in Figure 2, and there are no obvious macroscopic defects in WJ, such as solidification cracks or liquation cracks. The grain morphologies of HAZ are all equiaxial austenite grain with smooth grain boundaries. However, there is a significant grain coarsening in CGHAZ. As the distance from the fusion line increases, the grain size gradually decreases. The average grain size of the FL, FL + 2 mm locations and BM is around 62.0 μm, 22.1 μm and 21.2 μm, respectively (disregard twin boundary). Please note that the grain size of FL + 2 mm location is close to that of the BM, indicating that there is no significant grain growth at the FL+2 mm location.

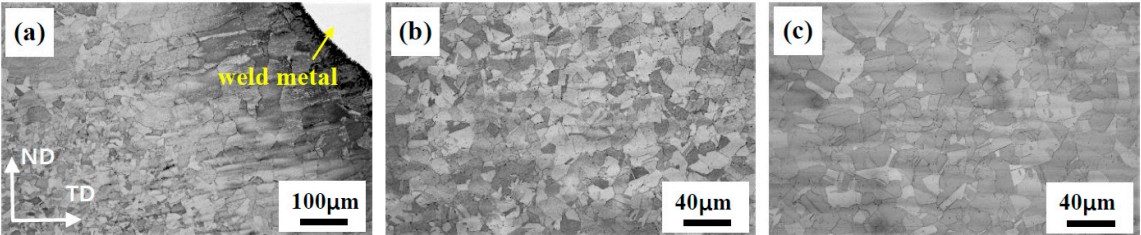

**Figure 2.** Optical microstructure of WJ at (**a**) FL, (**b**) FL + 2 mm location and (**c**) BM.

### 3.2. Tensile Properties at Room and Cryogenic Temperatures (−196 °C)

The engineering stress-strain curves of BM and WJ at room and cryogenic temperatures are shown in Figure 3a, and the detailed tensile properties are listed in Table 3. At room temperature, the yield strength of WJ increases slightly with respect to the BM, whereas the tensile elongation of WJ decreases

by ~57%. However, when the testing temperature is lowered to −196 °C, the yield strength of the BM and WJ is nearly twice as much as that at room temperature. Moreover, the tensile strength of the BM and WJ is greater than 1400 MPa and 1000 MPa, respectively. Please note that the tensile elongation of the WJ greatly decreases by ~60%, whereas the tensile elongation of the BM increases by ~25%. Regardless of testing temperatures, the WJ was broken at WM due to its poorer strain hardening capacity with respect to the BM.

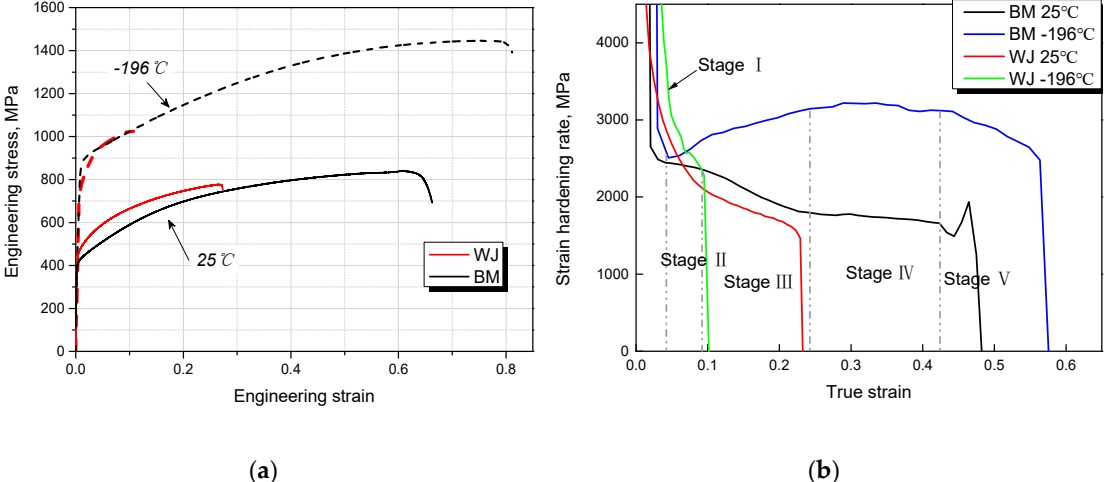

(**a**) (**b**)

**Figure 3.** (**a**) Engineering stress-strain curves, (**b**) strain hardening rate curves of WJ and BM at room temperature and −196 °C.

The strain hardening rate (SHR) curves of BM and WJ at room and cryogenic temperatures are shown in Figure 3b. In the early deformation stage, there is a steep decrease in SHR for both the WJ and the BM at room temperature and −196 °C owing to rapid dynamic recovery caused by cross slip and annihilation of dislocations [24], but the decrease rate of SHR of the WJ is relatively smaller than that of the BM. In addition, the SHR at −196 °C is greater than that at room temperature. Especially for the BM, the SHR can be greatly enhanced at −196 °C. At room temperature, the SHR curve of BM is divided into five stages. At stage I, there is a steep decrease in SHR. With further increasing strain, the SHR decreases slowly due to the initiation of primary twinning [25] at stage II. Please note that the transition point between stage I and stage II is considered to be the onset of mechanical twins [26,27], and this point is about 0.03 true strain at room temperature. I. Gutierrez et al. [28] demonstrated that the onset of mechanical twins is also 0.03 true strain in Fe-22Mn-0.6C TWinning Induced Plasticity (TWIP) steel by electron channeling contrast (ECCI). De Cooman et al. [25] also indicated the primary twinning stage stars at around 0.03–0.04 true strain. These results are comparable with our estimated result. At −196 °C, the SHR of BM is much higher than that at room temperature owing to more active mechanical twins at cryogenic temperature [29]. The initiation strain of primary twinning was estimated to be 0.045 according to the transition point between stage I and stage II. On the other hand, on the basis of the strains of the onset of mechanical twins at room temperature and −196 °C, the tensile stress for twinning were estimated to be around 480 MPa and 965 MPa, respectively, exhibiting a higher stress needed to activate mechanical twins at −196 °C.

**Table 3.** Tensile properties of BM and WJ at room temperature and −196 °C.

| Test Temperature | Steel | Yield Strength, MPa | Tensile Strength, MPa | Tensile Elongation, % |
|---|---|---|---|---|
| 25 °C | BM | 423 ± 16 | 853 ± 14 | 64.2 ± 1.4 |
| | WJ | 444 ± 5 | 768 ± 8 | 27.7 ± 2.1 |
| −196 °C | BM | 871 ± 3 | 1447 ± 2 | 80.1 ± 1.8 |
| | WJ | 804 ± 15 | 1027 ± 6 | 11.2 ± 1.3 |

### 3.3. Vickers Macrohardness Distribution

The Vickers macrohardness profile across the WJ is shown in Figure 4. It is evident that the macrohardness of HAZ decreases gradually with an increase in the distance from the fusion line. The Vickers macrohardness tends to be unchanged at distances higher than 18 mm. Please note that the macrohardness of HAZ is higher than that of BM, especially in CGHAZ. Hardening was observed as a result of grain coarsening in the HAZ of a Fe-18Mn-0.6C-1.5Al steel, decreasing from 275 HV to 220 HV [30]. An increased hardening occurred in the fusion zone of Fe-25Mn-3Al-3Si steel, which underwent grain refinement: the hardness increased from 180 HV to 250 HV [31]. For structural steels with body-centered cubic (BCC) crystal structure, the hardness of the HAZ would increase owing to the formation of quenching martensite. However, for austenitic steels, there is no phase transformation during post-weld cooling, grain size and chemical composition should be important factors affecting the hardness. In the HAZ, the microstructure exhibits no obvious grain refinement, whereas the macrohardness increases from ~202 HV in the BM to ~260 HV in the HAZ. Hence, it is reasonable to deduce that there exist some other factors hardening the austenite grain in the HAZ, which will be elaborated in the discussion section. In addition, the BM has the lowest macrohardness due to a lower yield strength. On the other hand, the macrohardness of the WM is slightly higher than that of BM.

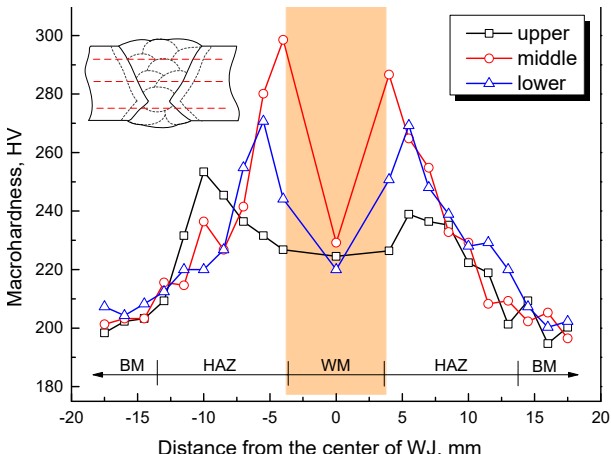

**Figure 4.** Vickers macrohardness profile across the WJ. There are three measuring lines across the WJ which are illustrated on the top left.

### 3.4. Cryogenic Charpy Impact Toughness

The cryogenic Charpy V-notch (CVN) impact absorbed energies at different locations of the WJ, as well as the corresponding crack propagation and fracture morphology, are shown in Figure 5. The cryogenic impact toughness at the WM and FL locations is worst, and both Charpy impact absorbed energies are about 55 J. According to standard in Reference [32], the cryogenic impact absorbed energy of the WJ meets the requirements for LNG tank building. Besides, the tensile properties of the WJ (as described in Section 3.2) are also qualified.

As the distance from the FL location increases, the cryogenic impact absorbed energies increase gradually. When this distance exceeds 5 mm, the cryogenic impact absorbed energies tend to be unchanged. Please note that FL is defined as the location where half of the V-notch is in WM, hence the decreased cryogenic impact absorbed energy at the range of FL to FL + 2 mm location is the resultant of WM and base metal. In addition, FL + 2 mm is all located at the base metal, therefore the decreased cryogenic impact absorbed energy at FL + 2mm location means that there is a deteriorated cryogenic impact toughness in the HAZ.

Figure 5b,c reveals secondary cracks underneath fracture surface at the FL location and BM. The secondary crack propagation path at FL location is nearly straight, but the secondary crack propagation path of the BM is ragged. Additionally, there are many mechanical twins close to the

secondary crack of the BM (indicated by the yellow arrows in Figure 5c), indicating that a very severe plastic deformation has occurred, thereby leading to a higher energy absorption. However, there are no obvious mechanical twins close to the secondary crack at the FL location. Furthermore, it can be found that both high angle grain boundaries (HAGBs) and twin boundaries (TBs) can hinder crack propagation (indicated by the orange arrows in Figure 5c). A higher impact absorbed energy can be hence obtained.

Fractographs of Charpy impact samples at the FL location and BM are presented in Figure 5d,e. Both fractographs exhibit typical ductile dimple fracture features, but the dimple size and depth of the BM are larger and deeper than that at the FL location. In addition, there are more facets in fractograph at the FL location. To recap, the deformation degree of austenite close to impact fracture surface at the FL location is much less than that of the BM.

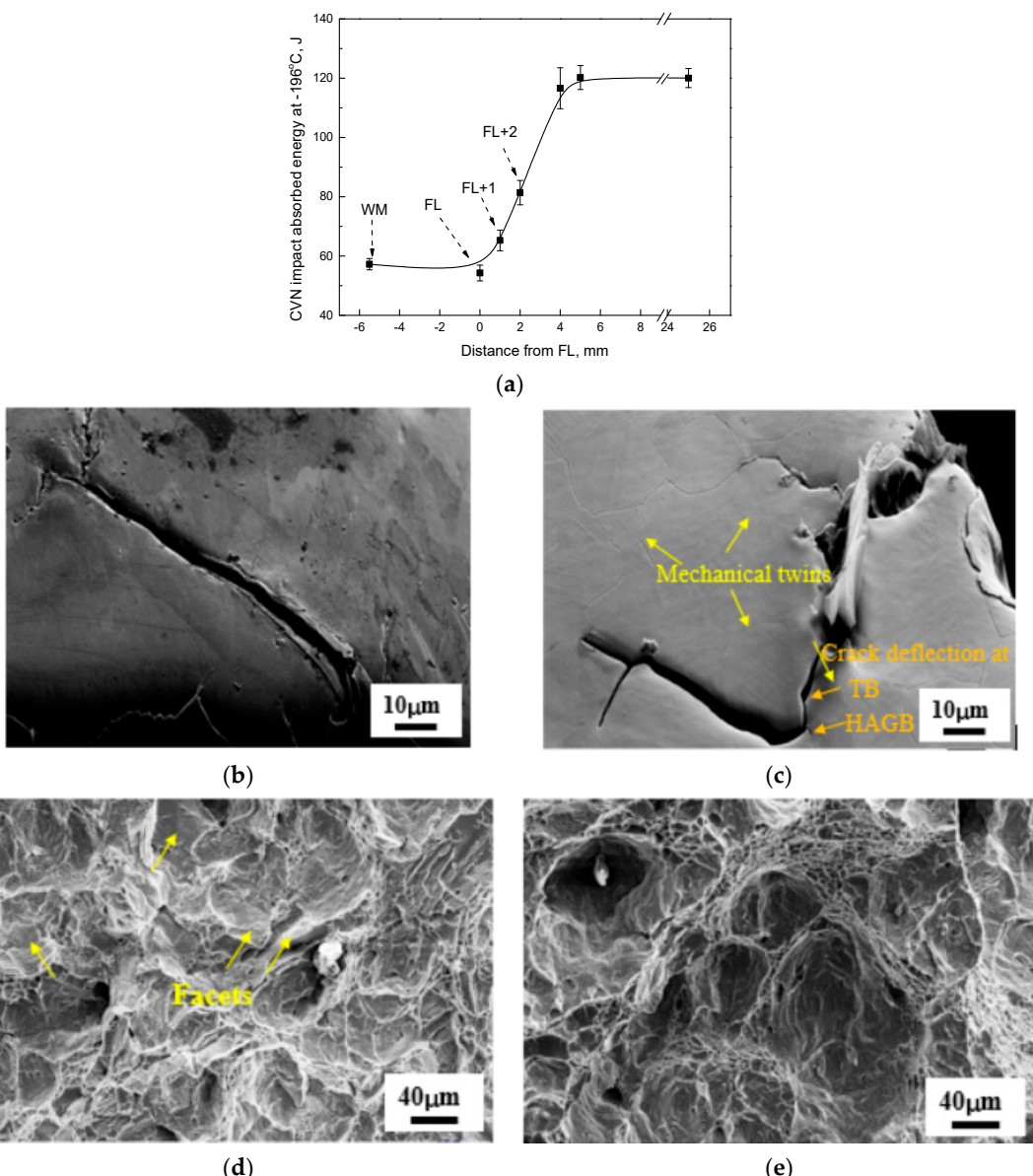

**Figure 5.** (**a**) Cryogenic Charpy V-notch impact absorbed energy profile across the WJ, secondary cracks underneath fracture surface at the (**b**) FL location, (**c**) BM, and fracture morphology of impact samples at the (**d**) FL location, (**e**) BM. The mechanical twins are indicated by yellow arrows in (**c**) and the crack deflection at TB and HAGB are also indicated by orange arrows in (**c**). The facets in fractograph are also indicated by yellow arrows in (**d**).

### 3.5. Deformation Behavior under Cryogenic-Temperature Impact Loading

It has been demonstrated that the cryogenic impact toughness of high-Mn austenitic steel has a strong relation to the twinning behavior [33,34]. We also observed that mechanical twins play an important role in crack nucleation and propagation (Figure 5c). To reveal the reasons for the deterioration of cryogenic impact toughness in CGHAZ, the twinning behavior of deformed microstructures underneath fracture surface at the FL location and BM was analyzed via TEM. Figure 6 shows the deformed microstructures close to fracture surface of Charpy V-notch impact samples at the FL location and BM, showing that nano-twins and dislocation substructure are the primary deformed microstructures. Moreover, multiple twinning systems were activated in the deformed grains of the two samples. However, there exists a significant difference in the mechanical twin thickness and inter-twin spacing between the two samples. The mechanical twin thickness and inter-twin spacing of the deformed microstructures close to fracture surface of Charpy V-notch impact samples at the FL location and BM were measured and the results are shown in Figure 7a,b. The results show that the average twin thicknesses of deformed microstructures at the FL location and BM were estimated to be around 35.2 nm and 28.2 nm, respectively, and the average inter-twin spacings at the FL location and BM were estimated to be around 183.2 nm and 98.8 nm, respectively. These results indicate that the deformed austinite grains of the BM are refined sufficiently by thinner mechanical twin thickness and smaller inter-twin spacing, which can efficiently reduce the mean free path of dislocation slip [35,36], thereby leading to enhanced plastic deformation capacity due to high strain hardening rate.

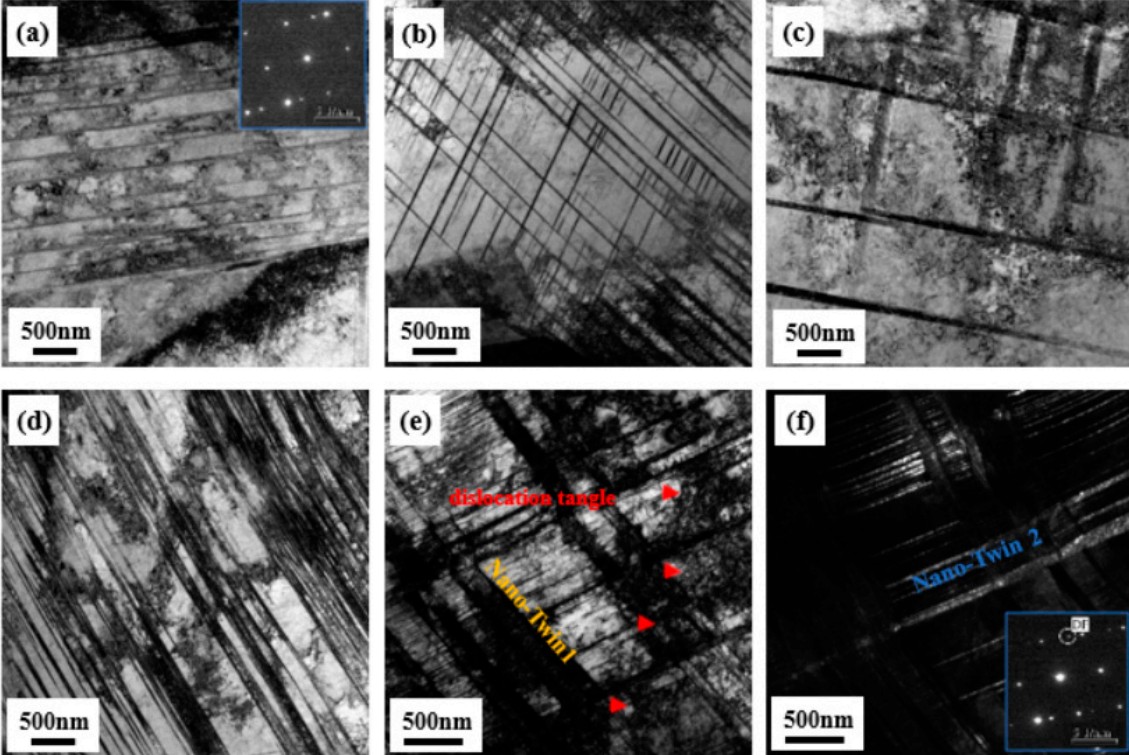

**Figure 6.** TEM micrographs of the deformed microstructures close to fracture surface of Charpy V-notch impact samples at the FL location and BM, (**a**–**c**) FL, (**d**–**f**) BM: (**a**) bright-field image showing nano-twins formed in primary twinning system and (**b**,**c**) bright-field images showing nano-twins formed in two twinning systems, (**d**) bright-field image showing nano-twins formed in primary twinning system, (**e**) bright-field image and (**f**) corresponding dark-field image showing nano-twins formed in two twinning systems.

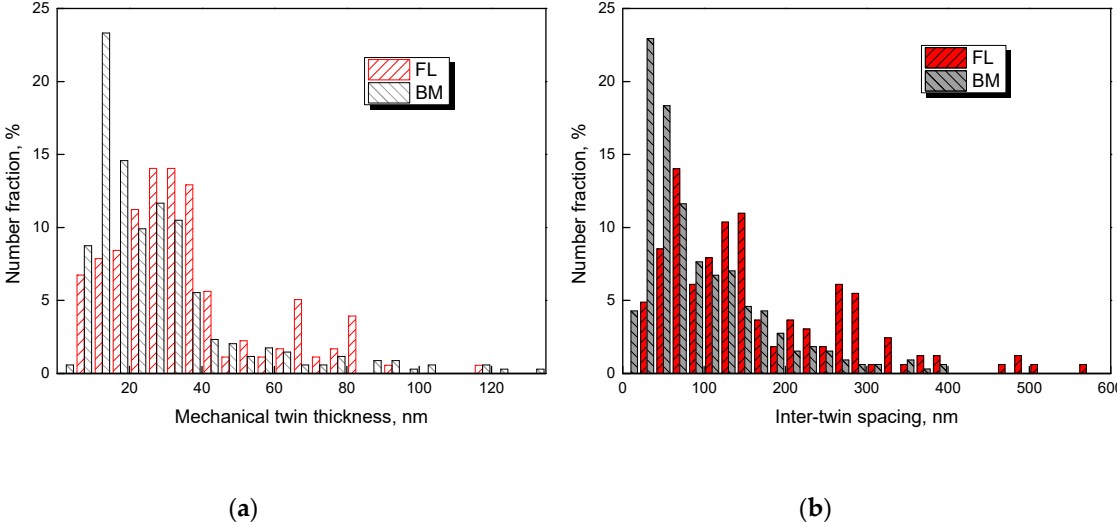

**Figure 7.** Distribution map of (**a**) mechanical twin thickness and (**b**) inter-twin spacing in deformed microstructures close to fracture surface of impact samples.

## 4. Discussion

There is a significant decrease in cryogenic impact absorbed energy in HAZ, especially in CGHAZ (at the FL location). The grain size in CGHAZ exhibits obvious grain coarsening with respect to the BM. In body centered cubic steels, grain refinement is an effective approach to improve the cryogenic impact toughness due to the resistance of grain boundary to crack [37]. Therefore, the grain coarsening will lead to an obvious deterioration of cryogenic impact toughness in CGHAZ of conventional structural steels with BCC crystal structure [38]. In high-Mn austenitic steel, in addition to the dislocation slip, twinning is also an important deformation mechanism. Consequently, the grain coarsening has two aspects affecting cryogenic impact toughness. On the one hand, grain coarsening lowers the fraction of grain boundary area, thereby leading to less energy absorption. On the other hand, larger grain size also lowers the critical resolved shear twinning stress ($\tau_T$) according to Equation (1) [28].

$$\tau_T = \frac{\gamma_{isf}}{b_p} + \frac{G \cdot b_p}{D} \tag{1}$$

where $\gamma_{isf}$ is the intrinsic SFE, $b_p$ is the Burgers vector of a Shockley partial dislocation, G is the shear modulus, and $D$ is the average grain size. Generally, a smaller $\tau_T$ indicates easy twinning. The activation of mechanical twins will lead to higher strain hardening capacity owing to the dynamic Hall-Petch effect, resulting in a higher impact absorbed energy. Therefore, there are two contradictory aspects concerning the effect of grain coarsening on the cryogenic impact toughness of high-Mn austenitic steel. Wang et al. [39] reported that the impact absorbed energy of a Ti–V–Mo-containing high-Mn steel increases from 84 J to 166 J as the grain size increases from 10.4 μm to 21.7 μm by solution treatment. Chen et al. [34] also demonstrated that the impact absorbed energy of a hot-rolled TWIP steel can be enhanced by ~26 J as the grain size increases from 8.5 μm to 17.3 μm without much sacrifice of yield strength. These results imply that the larger grain size is benefit to the cryogenic impact toughness of high-Mn austenitic steel. In other words, the positive effect of twinning exceeds the negative effect of decreased grain boundaries, leading to an increase of cryogenic impact absorbed energy. In the present study, the cryogenic Charpy V-notch impact absorbed energy vs. average grain size at different locations of HAZ was plotted in Figure 8. Please note that the CGHAZ in FL location possesses the largest grain size, whereas its cryogenic impact absorbed energy is the lowest, exhibiting a different result with earlier works. The poorer cryogenic impact toughness of the WM may be one aspect deteriorating the cryogenic impact toughness at the FL location. However, at the FL + 2 mm location where the notch is all located at the base metal, the microstructure having a larger grain size

still exhibits a decreased cryogenic impact absorbed energy with respect to the BM. Therefore, it is reasonable to deduce that the deteriorated cryogenic impact toughness in the HAZ is derived from other factors.

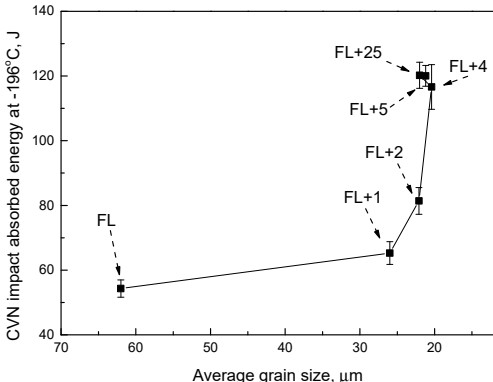

**Figure 8.** The correlation of cryogenic Charpy V-notch impact absorbed energy and average grain size at different locations of HAZ.

In addition to grain coarsening, severe alloying element segregation and depletion in CGHAZ is presented by the EPMA composition analysis, as shown in Figure 9. There is obvious band-like segregation of alloying elements such as Fe, Mn, C and Si, especially for Mn and C elements. Furthermore, the C element also has a pronounced segregation near the FL. On the one hand, the C-Mn-Si segregation band shows relatively high strength due to more interstitial carbon and silicon atoms. On the other hand, the rapid cooling of WJ during the welding presumably leads to high defect density in HAZ [11,40], and this result can be supported by higher local orientation gradients in the HAZ with respect to the BM (Figure 10). Therefore, the two abovementioned aspects lead to the increased macrohardness of HAZ. Meanwhile, the hardened austenite also deteriorates the plastic deformation capacity, leading to a decreased impact absorbed energy of CGHAZ.

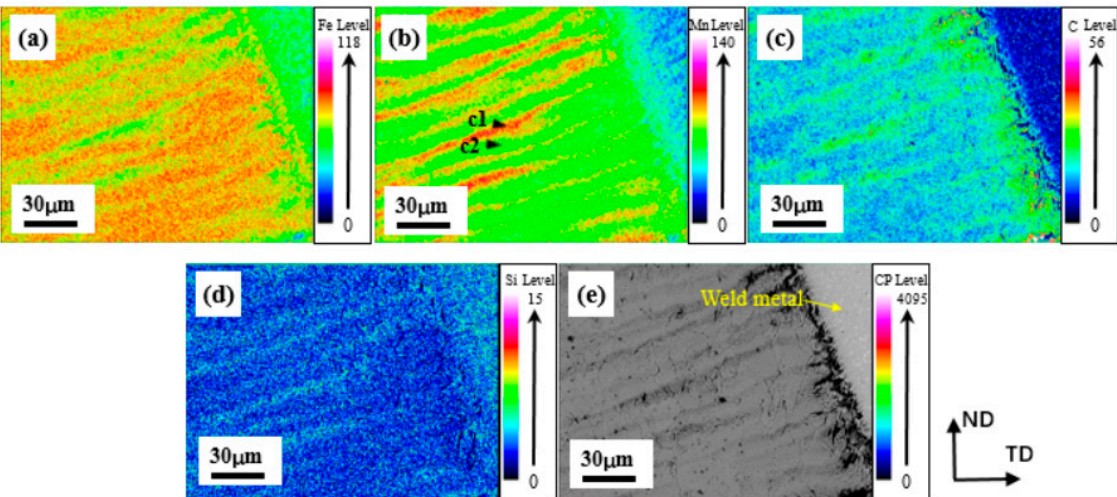

**Figure 9.** EMPA composition maps of (**a**) Fe, (**b**) Mn, (**c**) C, (**d**) Si elements and (**e**) EBSP in CGHAZ. The c1 and c2 arrows indicate the C-Mn-Si segregation band and C-Mn-Si depletion band, respectively.

However, the banded segregation behavior of Mn and Fe elements of the BM (Figure 11) is weaker, and the distribution of C and Si elements in austenitic matrix is homogeneous with respect to CGHAZ. Furthermore, the direction of Mn segregation band in the CGHAZ and the BM are different. In the BM, the orientation of Mn segregation band is parallel to the rolling surface; however, the Mn segregation

band is perpendicular to the fusion line in CGHAZ. Hence, it is reasonable to deduce that the formation of C-Mn-Si segregation band in CGHAZ is related to welding.

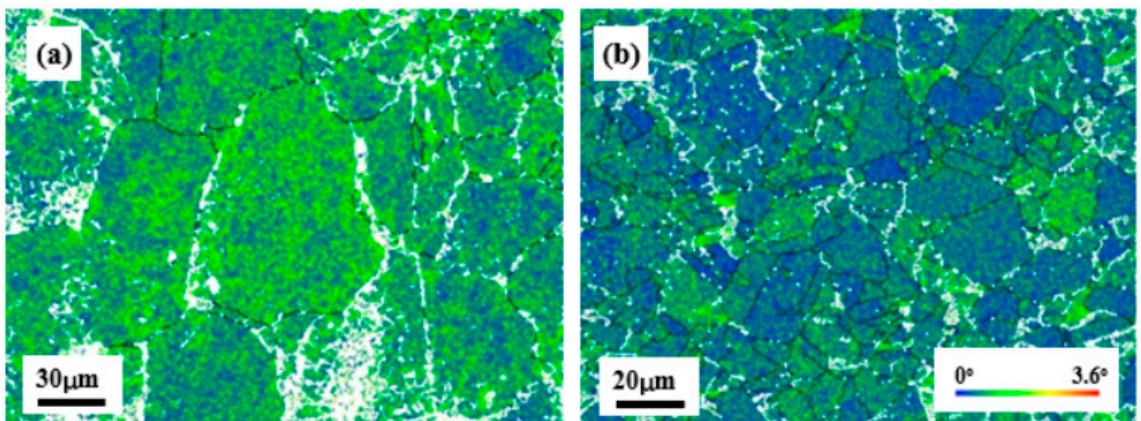

**Figure 10.** KAM maps, showing local orientation gradients in grain interior of (**a**) HAZ and (**b**) BM.

The SFE is affected by chemical composition and temperature [29,41,42]. Solute segregation can also lead to the change in local SFE, which in turn affects twinning behavior [33,43]. The analyses of deformation behavior (as described in Section 3.5) demonstrate that the BM has a more active twinning behavior with respect to the CGHAZ. Thereby, wavelength dispersive spectrometer (WDS) was applied to quantitatively analyze the chemical composition of the C-Mn-Si segregation band and C-Mn-Si depletion band in the CGHAZ and BM, and the selected micro zones for elemental composition analysis are indicated by the black arrows in Figures 9b and 11b. The analysis results of alloying elements are shown in Table 4.

From Equation (1), the $\tau_T$ depends on $\gamma_{isf}$ and grain size, Gutierrez-Urrutia et al. [28] also indicated that the contribution of SFE to the twinning stress is larger than that of the grain size when the grain size exceeds 3 μm. Hence, it is reasonable to deduce that the segregation of C, Mn, Si elements has an important influence on $\tau_T$ due to the different SFE. Thereby, the SFE and $\tau_T$ of the selected micro zones of CGHAZ and BM were estimated based on Equation (1) and the composition-temperature-related equation of $\gamma_{isf}$:

$$\gamma_{isf} = 2\rho\Delta G^{\gamma\rightarrow\varepsilon} + 2\sigma^{\gamma/\varepsilon} \tag{2}$$

where $\Delta G^{\gamma\rightarrow\varepsilon}$ is the Gibbs energy, which is the chemical force for the bulk $\gamma \rightarrow \varepsilon$ transformation, $\sigma^{\gamma/\varepsilon}$ is the $\gamma/\varepsilon$ interface energy, and its value is usually between 6 and 10 mJ/m$^2$ [6]. $\rho$ is the molar surface density along the {111} plane (in moles per unit area), its value can be determined by lattice constant $\alpha_\gamma$ (0.36 nm for TWIP steel), it follows that:

$$\rho = \frac{4}{\sqrt{3} \cdot \alpha_\gamma^2 \cdot N_A} \tag{3}$$

Assuming b = $2.5 \times 10^{-10}$ m, G = 65 GPa [28], and considering the results in Section 3.2 that the $\tau_T$ at −196 °C is about twice as much as that at 25 °C, the $\tau_T$ of micro zones in the CGHAZ and BM were calculated and the results are shown in Table 4. It is demonstrated that the C-Mn-Si segregation band of CGHAZ has a higher $\tau_T$ with respect to that of the BM. Additionally, the difference of $\tau_T$ between the C-Mn-Si segregation band and C-Mn-Si depletion band of the CGHAZ is much greater than that of BM, indicating the different homogeneity of twinning deformation.

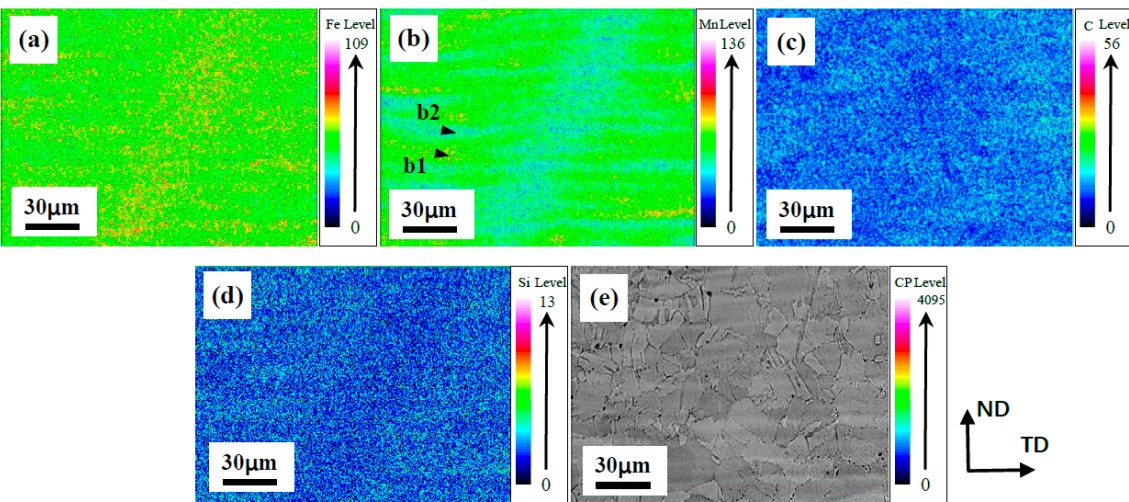

**Figure 11.** EMPA composition maps of (**a**) Fe, (**b**) Mn, (**c**) C, (**d**) Si elements and (**e**) EBSP in BM. The b1 and b2 arrows indicate the Mn segregation band and Mn depletion band, respectively.

The analyses above indicate that the reasons for the deteriorated cryogenic impact toughness of the CGHAZ can be summarized as three aspects. Firstly, the higher $\tau_T$ in the C-Mn-Si segregation band of the CGHAZ means difficultly twinning. Under cryogenic-temperature impact loading, the twin formation becomes more active with decreasing SFE in the TWIP range [44], whereas the migration of dislocations would become difficult due to the decreased temperature [45]. Hence, the contribution of twinning to plastic deformation becomes more important at cryogenic temperature. On the one hand, twinning could enhance the multiplication of dislocations by efficiently reducing the mean free path of dislocations [35], on the other hand, twin boundaries could also act as strong obstacles for dislocation gliding, leading a dynamic Hall-Petch effect [46]. Therefore, the inhibited twin formation in CGHAZ leads to a deteriorated plastic deformation capacity. Secondly, the inhomogeneous microstructure of CGHAZ means that local deformation occurs in the C-Mn-Si depletion band initially because of its lower strength and $\tau_T$, and the plastic deformation will be confined to these zones until their strength equal to that of the C-Mn-Si segregation band [14]. Then, the early micro voids would readily form in the low strength zone due to dislocation tanglement during plastic deformation [12], thereby leading to less energy absorption of CGHAZ. Thirdly, the higher defect density in hardened austenite would retard the migration of dislocations, leading to local stress concentration, and finally resulting in deteriorated cryogenic impact toughness of CGHAZ.

**Table 4.** WDS quantitative analysis results of detected micro zones and the corresponding $\gamma_{isf}$, $\tau_T$ of CGHAZ and BM.

| Locations | C, wt.% | Mn, wt.% | Si, wt.% | Al, wt.% | $\gamma_{isf}$ (25 °C)mJ/m$^2$ | $\tau_T$ (−196 °C), MPa |
|---|---|---|---|---|---|---|
| c1 | 0.81 | 28.8 | 0.50 | 1.56 | 54.2 | 434.1 |
| c2 | 0.44 | 22.6 | 0.34 | 1.63 | 35.0 | 280.5 |
| b1 | 0.61 | 26.2 | 0.50 | 1.53 | 45.2 | 371.1 |
| b2 | 0.53 | 23.8 | 0.44 | 1.54 | 39.5 | 317.4 |

## 5. Conclusions

A novel high-manganese cryogenic steel was successfully welded by multi-pass SMAW, and the tensile properties of WJ and resultant cryogenic impact toughness of HAZ were studied. The major conclusions are summarized as follows:

(1)  At room temperature, the WJ has the yield strength, tensile strength and tensile elongation of 444 MPa, 768 MPa and 27.7%, respectively. The yield strength of WJ increases by 21 MPa

with respect to the BM, but the tensile strength and elongation decrease obviously. At −196 °C, the yield strength of the BM and WJ is nearly twice as much as that at room temperature. The yield strength of WJ decreases slightly with respect to the BM, but the tensile strength and elongation decrease greatly. Regardless of testing temperatures, the WJ was broken at WM due to its poorer strain hardening capacity with respect to the BM.

(2) The cryogenic impact toughness at the WM and FL locations is the worst, and both Charpy impact absorbed energies are about 55 J. The fractographs of impact fractures in HAZ all have typical ductile dimple fracture features. However, the cryogenic impact toughness of CGHAZ is decreased, and the analyses shows that the twinning behavior in CGHAZ is inhibited with respect to the BM.

(3) There is severe C-Mn-Si alloying segregation and depletion in the CGHAZ. The higher $\tau_T$ in the C-Mn-Si segregation band of CGHAZ indicates difficulty twinning, which leads to a deterioration in plastic deformation capacity at cryogenic temperature. Additionally, the inhomogeneous microstructure of CGHAZ would promote the localized plastic deformation, readily leading to the formation of micro voids. Additionally, the higher defect density in hardened austenite also retards the migration of dislocations, finally deteriorating the cryogenic impact toughness of the CGHAZ.

**Author Contributions:** Z.-Y.L. and J.C. conceived the experiments; J.-K.R. and Q.-Y.C. performed the experiments; J.-K.R. wrote the paper; J.C. revised the paper and contributed to the discussion of the results. All authors have read and agreed to the published version of the manuscript.

**Funding:** This research was funded by the National Key R&D Program of China (2017YFB0305000), the Fundamental Research Funds for Central Universities (N170708018) and the APC was funded by China Postdoctoral Science Foundation (2018M630295, 2018T110229).

**Conflicts of Interest:** No conflict of interest exits in the submission of this manuscript, and manuscript is approved by all authors for publication. I would like to declare on behalf of my co-authors that the work described was original research that has not been published previously, and is not under consideration for publication elsewhere, in whole or in part. All the authors listed have approved the manuscript that is enclosed.

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
