# Peer review of "On Mechanical Properties of Welded Joint in Novel High-Mn Cryogenic Steel in Terms of Microstructural Evolution and Solute Segregation"

_metals, doi:10.3390/met10040478_

Round 1

Reviewer 1 Report

Dear Authors,

I have read Your submission with great attention. In my opinion, the work has the right structure, is well thought out in all respects, the tests are carried out correctly, and the conclusions I consider important. I am convinced that the research material described in the manuscript will be an important source of information for industrial engineers and scientists. While reading the article, I had a few comments that, hopefully, will help shape the final version of your article.

I write remarks in order, giving the line number:

48: remove “welding”.

51: abbreviation “HF” is not explained.

51 and 52: change “welded” to “welding” (2 times).

71: I propose to add the table with chemical composition of base materials and filler metal.

76: Please add designation (ISO or AWS) and diameter of electrode.

80: add “arc” before “voltage” and “welding” before “current”.

What was the width of the sample? I suggest you add a scheme showing where sampling takes place. Were there any rules, norms or recommendations when choosing the place where the sample was removed?

89: in liquid nitrogen?

95: please explain the abbreviation IGF.

115 change dot to space.

119 add space before A.

Figure 2: check the sharpness of the figure.

Figure 3a: it is necessary to thicken the dashed lines (for -196C), because the red line is hardly visible, which makes interpretation of the text from line 143 difficult.

167-170: change “Vichers” to “Vickers”.

Figure 4: Is "ave" necessary? In this case, does the average value have physical meaning or in relation to the application of this joint?

Format references consequently (titles).

Ref. 11: no title.

Ref. 30: correct the description for:Volume 737, Pages 158-165.

Ref. 35: should be: Taban, E., Deleu, E., Dhooge, A., & Kaluc, E. (2009). Laser welding of modified 12% Cr stainless steel: Strength, fatigue, toughness, microstructure and corrosion properties. Materials & Design, 30(4), 1193-1200.

Reviewer 2 Report

In this research work authors reported the microstructure evolution and mechanical properties of multi-pass shielded metal arc welding (SMAW). The current work provides very important information on the mechanical properties of welded joint such as tensile, macrohardness, and impact toughness under cryogenic environment compared to the base metal. Although there are lot of new findings were reported, however, the manuscript needs minor revision to be considered for publication to the journal of Metals. Some comments are listed below:

  1. In abstract, line 23-24, it is stated that “cryogenic toughness and tensile properties of the WJ meet the requirements for LNG tank building.” What is the required toughness and tensile properties for LNG tank? What standard or guideline is followed to determine the requirement?
  2. In page 2, lines 63-65, there is an ill constructed sentence like “The presence of electron probe microanalyzer (EPMA)….”. Correct it.
  3. Throughout the manuscript there are many typographical errors can be found, for example, Vichers. Figure 6 is missing.
  4. In page 5, lines 157-158, how the true strain for onset mechanical twins was determined? Same comment for true strain for primary twin formation.
  5. In lines 178-179, “In the HAZ, the grain size increases obviously with respect to the BM, whereas the macrohardness increases from ~202 HV in the BM to ~260 HV in the HAZ” the statement is misleading. It sounds like, the HAZ grain size is larger than BM; but that is not the case. Please correct.
  6. In page 9, lines 267-268, “These results imply that the larger grain size is benefit to the cryogenic impact toughness of high-Mn austenitic steel”, the above statement is true in general but nothing to do with CGHAZ as no attempt was taken to modify the CGHAZ grain size.
  7. It would be interesting to see impact toughness vs. grain size plot at different HAZ locations such as FL, FL+1, FL+2…..
  8. How does alloying segregation influence the defects density in HAZ?
  9. In line 301, “the red arrows in Figure 9e and Figure 11e”, there is no such red arrows on the Figures.
  10. Modify Figure 11 label from (a), (b), (c), (c), and (d) to (a), (b), (c), (d), and (e).
  11. How does the poor stain hardening capacity of WM leading to ductile dimple fracture?
